# Recent Advances in Clinical Research of Prophylactic Vaccines Against Tuberculosis

**DOI:** 10.3390/vaccines13090959

**Published:** 2025-09-10

**Authors:** Buyun Xu, Mengjuan Yuan, Lisa Yang, Lan Huang, Jingxin Li, Zhongming Tan

**Affiliations:** 1National Vaccine Innovation Platform, School of Public Health, Nanjing Medical University, Nanjing 210000, China; xby_9226@163.com (B.X.); yuanmj_ph@163.com (M.Y.); 18856984817@163.com (L.Y.); hl1815758708@163.com (L.H.); 2Jiangsu Provincial Medical Innovation Center, National Health Commission Key Laboratory of Enteric Pathogenic Microbiology, Jiangsu Provincial Center for Disease Control and Prevention, Nanjing 210000, China; 3School of Public Health, Southeast University, Nanjing 210000, China

**Keywords:** tuberculosis vaccine, safety, immunogenicity, clinical trials

## Abstract

Tuberculosis (TB), caused by Mycobacterium tuberculosis (MTB), is one of the leading infectious causes of adult mortality worldwide. The Bacillus Calmette–Guérin (BCG) vaccine is currently the only approved vaccine for TB prevention, but its protective efficacy against adult pulmonary TB is limited, and it lacks effective protection against primary or latent TB infection. There is an urgent need to develop more effective preventive TB vaccines. Currently, preventive TB vaccines under clinical investigation globally include live attenuated vaccines, recombinant subunit vaccines, viral vector vaccines, and mRNA vaccines. This article reviews and summarizes recent progress in the clinical development of preventive TB vaccines, analyzing and comparing their safety, immunogenicity, and protective efficacy. It also explores novel strategies for next-generation TB vaccine development, aiming to provide insights and directions for future research.

## 1. Introduction

Tuberculosis (TB), caused by Mycobacterium tuberculosis (MTB), is a chronic infectious disease that poses a severe threat to human health and remains one of the leading causes of adult mortality worldwide [1]. Although the global incidence of TB has been slowly declining with the advancement of antituberculosis drugs, challenges persist due to the emergence of drug-resistant MTB strains, the high prevalence of HIV, and global poverty [2]. According to the World Health Organization (WHO) 2024 Global Tuberculosis Report, in 2023, there were approximately 10.8 million new TB cases globally, resulting in 1.25 million deaths. China reported around 0.74 million new TB cases, ranking third worldwide [3]. Notably, the number of TB-related deaths in 2023 exceeded those from COVID-19, making TB the leading cause of death from a single infectious disease.

Although the Bacille Calmette–Guerin (BCG) vaccine provides significant protection against severe forms of TB in infants, such as disseminated TB and tuberculous meningitis [4], its protection against adult pulmonary TB and primary or latent TB infection is limited [5]. Consequently, BCG has a limited impact on reducing overall MTB transmission. Therefore, there is an urgent need to develop novel, safe, and effective TB vaccines to achieve the WHO target of reducing TB incidence by 80% by 2030 [3]. Currently, preventive TB vaccines in clinical development include live-attenuated vaccines, subunit vaccines, viral vector vaccines, and emerging mRNA vaccines [3].

This review aims to systematically summarize the latest research progress of preventive TB vaccines under development globally, analyze the characteristics of different types of preventive TB vaccines, and discuss novel approaches to TB vaccine development, with the hope of providing references for further research in this field.

## 2. Live Attenuated Vaccines

### 2.1. VPM1002

VPM1002 is a live attenuated recombinant MTB vaccine co-developed by Germany’s Vakzine Projekt Management GmbH and India’s Serum Institute of India Pvt. Ltd. (SIIPL). Its construction involved substituting the urease C gene (ureC) in the BCG genome with the listeriolysin O gene (hly) from Listeria monocytogenes [6]. This genetic alteration facilitates enhanced delivery of antigens and DNA into the host cell cytosol, which may improve the vaccine’s immunogenic properties.

An open-label, randomized Phase IIa trial was carried out in a high-tuberculosis prevalence area of South Africa. The study enrolled 48 HIV-uninfected newborns who had not received BCG, aiming to compare the safety and immunogenicity of VPM1002 with BCG. Findings revealed that adverse reaction rates were comparable between the VPM1002 group (25.82%, 95% CI: 21.91–30.21%) and the BCG group (25.97%, 95% CI: 23.67–28.43%) [7]. Notably, the incidence of injection site abscess formation was significantly lower in the VPM1002 group than in the BCG group (11.1% vs. 41.7%, *p* < 0.0321). Additionally, VPM1002 induced levels of multifunctional CD4 + and CD8 + T cells similar to those observed with BCG, demonstrating that VPM1002 is safe, immunogenic, and well-tolerated in newborns.

A subsequent double-blind, randomized, controlled Phase IIb trial in South Africa further evaluated VPM1002 against BCG, including both HIV-infected and uninfected newborns. A total of 416 infants were randomized in a 3:1 ratio to receive VPM1002 or BCG [8]. The results showed that grade 3–4 vaccine-related adverse reactions or lymphadenopathy ≥10 mm occurred in 2% (7/312) of the VPM1002 group versus 33% (34/104) of the BCG group (risk difference: −30.45%, 95% CI: −39.61% to −21.28%), confirming the non-inferior safety profile of VPM1002 relative to BCG. However, at Week 6, the VPM1002 group exhibited significantly lower levels of IFN-γ in supernatants of purified protein derivative (PPD)-restimulated blood samples (−190.02 pg/mL, 95% CI: −344.23 pg/mL to −67.79 pg/mL, *p* < 0.0005) and a lower percentage of multifunctional CD4 + T cells (−0.184%, 95% CI: −0.253% to −0.114%, *p* < 0.001) compared to the BCG group.

Currently, a multicenter Phase III trial (NCT04351685, *n* = 6940) is ongoing in sub-Saharan Africa, with completion expected in 2025, to assess VPM1002’s efficacy, safety, and immunogenicity versus BCG in preventing MTB infection in newborns. Additionally, a Phase II/III trial (NCT03152903) in India is evaluating VPM1002’s ability to prevent tuberculosis recurrence in cured adult patients.

### 2.2. MTBVAC

MTBVAC, a collaborative development by Spain’s Biofabri, University of Zaragoza, India’s Bharat Biotech, IAVI, and TBVI, represents another live attenuated MTB vaccine. It is distinctive as the sole live attenuated vaccine in clinical development derived from a genetically modified human MTB isolate [9,10]. Attenuation was achieved through deletion of the phoP and fadD26 virulence genes [11,12].

A Phase Ia double-blind, randomized, controlled trial involving 36 healthy Swiss adults (18–45 years) compared MTBVAC at doses of 5 × 10^3^, 5 × 10^4^, and 5 × 10^5^ CFUs with BCG at 5 × 10^5^ CFUs (3:1 randomization) [13]. The trial demonstrated comparable safety across all dose groups, with no vaccine-related serious adverse events (SAEs) reported.

A Phase Ib randomized, double-blind, controlled, dose-escalation study was conducted in 36 neonates in a high-TB-burden region of South Africa. This trial compared three MTBVAC doses (2.5 × 10^3^, 2.5 × 10^4^, and 2.5 × 10^5^ CFUs) with BCG at 2.5 × 10^5^ CFUs [14]. MTBVAC induced dose-dependent Th1 responses mediated by multifunctional CD4 + T cells. The highest-dose MTBVAC group (2.5 × 10^5^ CFUs) showed significantly higher frequencies of specific CD4 + T cells than the equivalent BCG dose at day 70 (*p* = 0.0085) and day 360 (*p* = 0.026). In contrast, the low-dose MTBVAC group (2.5 × 10^3^ CFUs) exhibited significantly lower specific CD4 + T cell frequencies than both BCG and the high-dose MTBVAC group at days 70, 180, and 360.

A potential limitation is that MTBVAC’s ESAT6-CF10 antigens may cross-react with interferon-gamma release assay (IGRA) diagnostics [15], highlighting the need for alternative diagnostic tools to avoid false-positive results for latent tuberculosis infection (LTBI) following vaccination.

Two ongoing clinical trials in South Africa, NCT02933281 and NCT03536117, are investigating various aspects of MTBVAC. NCT02933281 evaluates the safety and immunogenicity of four MTBVAC doses (5 × 10^3^, 5 × 10^4^, and 5 × 10^5^ CFUs) in adults with and without LTBI. NCT03536117 assesses three MTBVAC doses (2.5 × 10^4^, 2.5 × 10^5^, and 2.5 × 10^6^ CFUs) in newborns. Additionally, a randomized, double-blind, BCG-controlled, multicenter Phase III trial (NCT04975178) is underway in sub-Saharan Africa to evaluate MTBVAC’s efficacy, safety, and immunogenicity in HIV-exposed/unexposed, HIV-negative infants. A recent Phase IIb trial (NCT06272812) has also initiated to assess MTBVAC’s efficacy in preventing tuberculosis disease in LTBI-positive adolescents and adults (14–45 years). Results from these trials are awaited.

## 3. Recombinant Subunit Vaccines

### 3.1. M72/AS01E

Developed by GSK, M72/AS01E consists of the highly immunogenic MTB fusion protein M72 (Mtb39A + Mtb32A) combined with the AS01 adjuvant system. Early Phase I/II trials demonstrated good safety and immunogenicity in diverse populations [16,17,18,19].

A Phase IIb trial in South Africa, Kenya, and Zambia enrolled 3575 HIV-negative, IGRA-positive adults (18–50 years) with LTBI, randomized 1:1 to receive two doses of M72/AS01E or placebo at 0 and 30 days. Interim analysis showed 22 cases of microbiologically confirmed pulmonary TB in the placebo group (*n* = 1787) vs. 10 cases in the vaccine group (*n* = 1786), yielding a vaccine efficacy (VE) against progression to active TB of 54% (95% CI: 2.9–78.2%) [20]. Solicited AEs within 30 days post-injection were more frequent in the vaccine group (67.4% vs. 45.4%, RR = 1.48, 95% CI: 1.35–1.62), primarily due to injection site reactions and flu-like symptoms. Final 3-year analysis showed sustained efficacy of 49.7% (95% CI: 2.1–74.2%). Anti-M72 IgG geometric mean titers (GMT) peaked at 1:547 at month 2 post-vaccination and remained positive throughout follow-up. Frequencies of M72-specific CD4 + T cells increased markedly post-vaccination and persisted at month 36 without waning [21]. M72/AS01E is the only vaccine with a positive efficacy readout so far, representing an unprecedented outcome in decades of tuberculosis vaccine research and holding substantial clinical significance.

A Phase III randomized, placebo-controlled trial (NCT04556981) is assessing safety and immunogenicity in HIV-positive individuals (on treatment) in South Africa. Furthermore, a large Phase III efficacy trial (NCT06062238), supported by the Bill & Melinda Gates Foundation and Wellcome, plans to enroll 20,000 people living with HIV in South Africa. If successful, M72/AS01E could become the first new TB vaccine in over a century to prevent TB disease in adolescents and adults [22].

### 3.2. GamTBvac Vaccine

GamTBvac is a subunit vaccine candidate consisting of dextran-binding domain (DBD)-modified Ag85A and ESAT6-CFP10 MTB antigens, adjuvanted with DEAE-dextran 500 kDa and CpG oligodeoxynucleotide (ODN) [23].

A Phase I open-label clinical trial conducted in Russia recruited 60 healthy adults aged 18–49 who had previously received the BCG vaccine. The participants were randomly assigned in a 1:1:1:1:1 ratio to five groups receiving either a placebo or the Gam TBVac vaccine at doses of 1/4, 1/2, or full dose (25.0 µg DBD-ESAT6-CFP10 and 25.0 µg DBD-Ag85a), following a two-dose immunization schedule on days 0 and 57 [24]. The trial aimed to evaluate safety across all groups and immunogenicity in the two-dose groups. Over a 140-day observation period, 60 adverse events were recorded, all of which were mild. Among these, only two events (elevated C-reactive protein to 23 mg/L and fever up to 38.0°C) were potentially related to the vaccine, both occurring in the full-dose group (2/12, 16.7%). All adverse reactions were transient and resolved without additional treatment, indicating acceptable safety profiles for the vaccine candidate.

Immunogenicity was assessed using IGRA to measure IFN-γ concentration and ELISA to determine IgG titers. On day 42, the half-dose group demonstrated significantly higher IFN-γ concentrations (*p* = 0.0122) and specific IgG antibody titers (*p* < 0.01) compared to the 1/4-dose group, with evidence of sustained immune responses. Based on these findings, the half-dose group (0.5 mL) is recommended for further research and development.

A subsequent Phase II double-blind, randomized, multicenter, placebo-controlled trial enrolled 180 BCG-vaccinated healthy adults (18–49 years), randomized 3:1 to two 0.5 mL doses of GamTBvac or placebo at 0 and 57 days. Total AEs were significantly higher in the vaccine group (90.4% vs. 53.3%, *p* < 0.001) [25] but were mostly mild/moderate. One severe AE occurred (aspartate aminotransferase elevation to 686 U/mL, resolved in 2 weeks), with no SAEs. After DBD-Ag85a stimulation, IFN-γ levels peaked at day 21 (*p* < 0.0001); significant differences (*p* < 0.0001) were seen between vaccine and placebo groups after DBD-ESAT6-CFP10 stimulation. By day 78, 98% and 94% of vaccinees showed increased IgG against DBD-ESAT6-CFP10 and DBD-Ag85A, respectively (*p* < 0.01), confirming robust immunogenicity supporting efficacy studies.

In 2021, the Phase III clinical trial (NCT04975737) for the Gam TBVac vaccine has also been launched in Russia. The trial aims to evaluate the safety and efficacy of the Gam TBVac vaccine in preventing primary respiratory tuberculosis in healthy individuals aged 18 to 45 who are not infected with HIV. Currently, the trial is recruiting participants, and its results will determine whether the Gam TBVac vaccine has potential for tuberculosis prevention.

### 3.3. ID93 + GLA-SE Vaccine

The ID93 + GLA-SE vaccine is composed of ID93, which contains four antigens (Rv2608, Rv3619, Rv3620, and Rv1813) associated with MTB virulence or latency, combined with the GLA-SE adjuvant [26]. In a Phase IIa, randomized, double-blind, placebo-controlled clinical trial conducted in South Korea, 123 healthy healthcare workers aged 19–65 years who had previously received BCG vaccination were randomly assigned to receive three doses of 2 µg ID93 + 5 µg GLA-SE, 10 µg ID93 + 5 µg GLA-SE, or placebo, administered at days 0, 28, and 56. The results showed that 26 participants (24.3%) in the vaccine group reported 43 adverse events, with the most common being local injection reactions. No severe adverse events related to the vaccine were observed. The geometric mean titer (GMT) of antigen-specific IgG antibodies in the vaccine group was significantly higher than in the placebo group after the first vaccination (*p* < 0.001) and reached its peak 28 days after the third dose (2 µg ID93 + 5 µg GLA-SE group: 108,339.22, 95% CI: 75,498.62–155,464.92; 10 µg ID93 + 5 µg GLA-SE group: 124,120.57, 95% CI: 99,951.92–154,133.26). Additionally, the frequency of CD4 + T cells increased significantly in both dose groups (*p* < 0.05). One year after the final vaccination, the CD4 + T cell frequency in the 10 µg ID93 + 5 µg GLA-SE group was higher than in the 2 µg ID93 + 5 µg GLA-SE group (*p* = 0.0116). These findings indicate that the ID93 + GLA-SE vaccine induces strong antigen-specific cellular and humoral immune responses with acceptable safety [27].

Currently, the United Kingdom is preparing to conduct a Phase I clinical trial (NCT06670755) among healthy adults aged 18–55 years to evaluate the safety and immunogenicity of the ID93 + GLA-SE vaccine in individuals who have or have not received BCG vaccination. Notably, a Phase I clinical trial (NCT06714513) targeting healthy and stable older adults aged 55–74 years is ongoing. This trial aims to address the urgent need for effective tuberculosis prevention in aging populations, particularly in high-risk groups (Table 1).

### 3.4. H107e/CAF10b and AEC/BC02 Vaccines

The H107e/CAF10b vaccine is composed of eight recombinant fusion proteins, including PPE68, ESAT-6, EsPI (deleted aa 75–294), EspC, EspA, MPT64, MPT70, and MPT83, combined with the CAF^®^10b adjuvant as a subunit vaccine [28]. Currently, dose-finding and open-label Phase I clinical trials (NCT06050356) are being conducted in South Africa. The study aims to enroll 140 healthy adults aged 18–45 years to assess the safety, reactogenicity, and immunogenicity of the new vaccine and its components. The trial is expected to be completed by 2026.

The AEC/BC02 vaccine, developed by Anhui Zhifei Longcom Biopharmaceutical Co., Ltd., is composed of Mycobacterium tuberculosis (MTB) antigens Ag85B, ESAT6-CFP10, and adjuvant BC02 [29]. It is the first recombinant protein-based tuberculosis vaccine in China to enter clinical trials. The vaccine has completed Phase Ia and Ib clinical trials (NCT03026972, NCT04239313); however, the results are not yet publicly available (Table 1).

## 4. Viral Vector Vaccines

### 4.1. ChAdOx1.85A + MVA85A Vaccine

The MVA85A vaccine, developed by the University of Oxford, is a recombinant poxvirus vector vaccine based on the Ankara strain of vaccinia virus to express the Mycobacterium tuberculosis (MTB) antigen 85A and induce T-cell immune responses [30]. ChAdOx1.85A is another tuberculosis vaccine based on a chimpanzee adenovirus vector, expressing the same antigen as MVA85A [31]. Currently, a combination immunization strategy using ChAdOx1.85A for primary immunization and MVA85A for booster immunization is being investigated for the prevention of tuberculosis.

In 2016, a Phase I clinical trial was completed in the United Kingdom, involving 42 healthy adults aged 18–55 years who had previously received bacillus Calmette-Guérin (BCG) vaccination. The trial evaluated intramuscular administration of ChAdOx1.85A and included four groups: a sentinel group (*n* = 6; ChAdOx1.85A), Group A (*n* = 12; ChAdOx1.85A), Group B (*n* = 12; ChAdOx1.85A priming followed by MVA85A boosting at days 0 and 56), and Group C (*n* = 12; ChAdOx1.85A priming at days 0 and 28, followed by MVA85A boosting at day 119). The study observed that adverse reactions were mostly mild to moderate, with no serious adverse events reported, indicating good vaccine safety [32]. MVA85A boosting significantly enhanced the Ag85A-specific IFN-γ concentrations (*p* = 0.001 at day 63; *p* = 0.003 at day 126) and substantially increased the frequencies of antigen-specific CD4 + and CD8 + T cells following vaccination. These findings suggest that the ChAdOx1.85A + MVA85A prime-boost strategy warrants further investigation.

Currently, a Phase IIa clinical trial (NCT03681860) evaluating intramuscular administration of ChAdOx1.85A and MVA85A in adults and adolescents is completed in Uganda, but the results are not yet publicly available.

### 4.2. Ad5Ag85A Vaccine

The Ad5Ag85A vaccine, developed by McMaster University in Canada, is a recombinant human adenovirus type 5 (Ad5)-based tuberculosis vaccine that expresses MTB Ag85A antigen. It is designed for booster immunization following BCG vaccination [33].

In 2021, a Phase I clinical trial was completed in Canada, evaluating intranasal administration of the Ad5Ag85A vaccine in healthy adults who had previously received BCG vaccination. A total of 36 volunteers were randomly assigned in a 1:1:1 ratio to three groups: low-dose intranasal (1 × 10^6^ PFU), high-dose intranasal (2 × 10^6^ PFU), or intramuscular injection (1 × 10^8^ PFU) of Ad5Ag85A [34]. The results demonstrated that the vaccine was well-tolerated with no Grade 3–4 adverse events or serious adverse events reported. All three groups induced significant cellular immune responses. At week 2, cytokine concentrations, including IFN-γ (*p* = 0.002 for the low-dose group; *p* = 0.0049 for the high-dose group), TNF-α (*p* = 0.001 for the low-dose group; *p* = 0.002 for the high-dose group), and IL-2 (*p* = 0.002 for the low-dose group; *p* = 0.0068 for the high-dose group), were significantly increased in the intranasal groups. Additionally, the low-dose intranasal group exhibited a significant increase in CD4 + (*p* = 0.002) and CD8 + T cell frequencies (*p* = 0.0137). These findings indicate that the Ad5Ag85A vaccine can induce mucosal immune responses in the respiratory tract, with good immunogenicity, particularly in the low-dose intranasal group, warranting further investigation.

### 4.3. TB/FLU-05E Vaccine

The TB/FLU-05E vaccine, developed by the Smorodintsev Institute of Influenza in Russia, is designed to express MTB antigens TB10.4 and HspX, combined with the attenuated influenza strain A/PR8/NS124-TB10.4-2A-HspX [35]. Currently, a randomized, double-blind, placebo-controlled Phase I clinical trial (NCT05945498) has been completed in Russia to assess the safety, reactogenicity, and immunogenicity of a single intranasal dose of TB/FLU-05E in volunteers aged 18–50 years who have previously received BCG vaccination. The results of this trial are pending (Table 1).

## 5. mRNA Vaccines

The success of mRNA technology in COVID-19 vaccines has spurred interest in TB mRNA vaccines. BNT164 is the first TB mRNA vaccine candidate to enter clinical trials. A Phase Ia randomized, placebo-controlled, double-blind, dose-escalation trial (NCT05537038) is evaluating the safety and immunogenicity of BNT164 (BNT164a1 and BNT164b1) at three dose levels in healthy German adults (18–55 years). A Phase Ib/IIa randomized, placebo-controlled, observer-blind, dose-finding trial (NCT05547464) aims to assess BNT164 in BCG-vaccinated HIV-negative individuals and people living with HIV (Table 1).

## 6. Novel Strategies for Next-Generation TB Vaccine Development

### 6.1. Optimizing Antigen Selection

The MTB genome encodes approximately 4000 unique proteins; however, only 11 candidate vaccine antigens have entered clinical evaluation, and all are developed as fusion proteins [36]. These antigens primarily originate from the secreted protein families Ag85 and ESAT-6. Experimental studies in mice have shown that Ag85B is predominantly expressed during the early stages of MTB replication, and Ag85B-specific T cells are limited due to reduced antigen expression. In contrast, ESAT-6-specific T cells experience functional exhaustion due to prolonged antigen stimulation [37]. Additionally, research indicates that individuals with LTBI exhibit stronger IFN-γ responses to more latent antigens compared to active tuberculosis patients [38]. Therefore, when selecting candidate antigens, it is crucial to consider the target population of the vaccine—healthy individuals or LTBI populations—and to choose antigens expressed during the latent or proliferative stages of MTB accordingly.

Currently, a multi-stage vaccine platform based on peptides has been designed and characterized [39]. This platform includes CD4 + and CD8 + T cell epitopes capable of inducing relevant T cell responses to MTB, enabling the selection of antigens expressed during both the latent and active phases of MTB infection. Furthermore, researchers have identified a mycobacterial cell wall lipid, 6-methylmycolate (TMM), as a CD1b-presented T cell antigen. TMM exhibits dual roles in stimulating both innate and adaptive immunity [40].

In the future, advancements in bioinformatics, pangenomics, and structural vaccinology are expected to facilitate systematic screening for novel protective antigens in MTB, providing a foundation for the development of highly effective vaccine candidates.

### 6.2. Developing Novel Adjuvants

Globally, widely used aluminum adjuvants are insufficient to meet the demands of innovative vaccine development, particularly for recombinant subunit vaccines. Among tuberculosis vaccine candidates in clinical trials, adjuvants such as the LR-9 agonist CpG-ODN, lipid formulations, and emulsions (e.g., AS01, GLA-SE) have been predominantly utilized.

Currently, several novel adjuvants, including Advax™ (Delta inulin particles), PLGA, Bacillus subtilis spores, chitosan and its derivatives, PolyI:C, cyclic dinucleotides, glucans, immunostimulating complexes (ISCOMs), Lipokel (PamCys2 and 3NTA), nanoemulsions, and yellow Brazilian palm wax nanoparticles combined with HBHA protein, are under preclinical evaluation [41,42]. While the development of new adjuvants faces challenges, such as the potential risks of inducing autoimmune diseases or excessive inflammatory responses, and some candidates have stalled due to issues like instability or low immunogenicity, the critical role of adjuvants in enhancing the efficacy of subunit vaccines cannot be overstated. Future research must thoroughly validate the safety and efficacy of novel adjuvants through rigorous preclinical and clinical trials to achieve breakthrough applications in tuberculosis vaccine development [43].

### 6.3. Defining Clinical Trial Endpoints

TB vaccines are categorized into those aimed at preventing disease and those targeting infection. Most candidate vaccines in the clinical development pipeline are designed to prevent TB, but due to the significantly lower incidence of TB compared to MTB infection, clinical trials targeting disease prevention require at least 3 years or more to monitor TB cases following vaccination, incurring higher costs [44]. Some researchers consider developing vaccines with shorter timelines and lower costs for infection prevention in Phase II trials, to better understand the mechanisms of vaccine efficacy in humans and provide a platform for subsequent trials to select lead candidates [45]. However, a major challenge remains the lack of standardized tests to directly measure the occurrence, persistence, and clearance of asymptomatic MTB infections.

Additionally, in clinical trials targeting TB infection prevention, vaccines such as MTBVAC and GamTBvac, which express the ESAT-6/CF-10 antigens, may cause cross-reactivity with IGRA used for diagnostic testing, potentially leading to false-positive results [15]. Therefore, the development of new diagnostic technologies is necessary to obtain more accurate clinical endpoint results for TB infection. Furthermore, the selection of clinical trial endpoints must comprehensively consider regional disease burden and population MTB infection status to evaluate the public health significance of different endpoints.

### 6.4. Expanding Vaccine Efficacy Evaluation Metrics

Currently, in most clinical-stage vaccine studies, vaccine-induced protective-related biomarkers are limited to immunological markers [46], particularly T cell responses producing IFN-γ, multifunctional T cell responses, and antibody titers against MTB antigens. However, research has revealed that the cytokine co-expression profiles of memory T cells induced by TB vaccine candidates such as MVA85A, M72/AS01E, and ID93 + GLA-SE are highly similar [47], indicating the limited diversity of current TB vaccine candidates in terms of their characteristics. Therefore, there is an urgent need to develop new diagnostic technologies and immunological biomarkers, and to strengthen research on humoral immunity and metabolic indicators related to TB protection, in order to establish alternative indicators for evaluating vaccine efficacy.

Recent studies have shown that purified specific IgG from LTBI individuals can enhance intracellular killing of MTB and macrophage activation [48]. Additionally, B cells are components of protective granulomas and play a role in controlling MTB infection and preventing reactivation through multiple mechanisms [49]. Therefore, incorporating B cell epitopes into vaccine antigen design may promote a more balanced cellular and humoral immune response, leading to improved immune responses.

Furthermore, a study on biomarkers for predicting active TB [50] analyzed cohorts of household contacts of TB index cases and strict non-human primate challenge models to assess whether the integration of blood transcriptomes and serum metabolomes could improve predictions of TB progression. This study identified novel immunometabolic features associated with TB progression, including cortisol, tryptophan, glutathione, and tRNA charging networks. Currently, many biomarker identification techniques, including transcriptomics, proteomics, and metabolomics, are being used individually or in combination to identify potential biomolecular signatures and various combinations of biomarkers.

## 7. Conclusions

TB vaccine research is advancing through diverse strategies and novel technologies, including varied immunization routes, new adjuvants, and mRNA platforms. Despite challenges such as antigen optimization, endpoint definition, and limited evaluation metrics, multidisciplinary collaboration and advances like deep learning offer hope. The successful development of new vaccines is crucial for achieving global TB control and elimination goals. Sustained global efforts, supported by governments and international organizations, are essential to foster the necessary research and collaboration.

## Figures and Tables

**Table 1 vaccines-13-00959-t001:** Essential Information on TB Vaccines Under Clinical Investigation.

Vaccine	NCT Number	Phase	Start Time	End Time (Estimated)	Population	Sample Size	Administration Route	Evaluation/Observation Indicators	Results
**VPM1002**	NCT01479972	II	2011.11	2012.11	Newborn infants	48	i.d.	1. Safety 2. Immunogenicity	Inoculation with VPM1002 can induce multifunctional CD4 + and CD8 + T cells
	NCT02391415	II	2015.6	2017.11	Newborn infants	416	i.d.	The difference between the VPM1002 and BCG vaccination groups in the incidence of grade 3 and 4 adverse drug reactions and IMP-related ipsilateral or generalized lymphadenopathy of 10 mm or greater (diameter)	VPM1002 is safe for both HIV-exposed and unexposed infants; both VPM1002 and BCG have immunogenicity, and from the 6th week onwards, the immune response intensity induced by BCG is greater than that of VPM1002
	NCT03152903	III	2017.12	2024.7	Healthy adults	2000	i.d.	Percentage of bacteriologically confirmed TB recurrence cases	NA
	NCT04351685	III	2020.11	2025.12	Newborn infants	6940	i.d.	Incident cases of QFT conversion	NA
**MTBVAC**	NCT02013245	I	2013.1	2014.11	Healthy adults	34	i.d.	Number of participants with AEs up to 210 days after vaccination	MTBVAC exhibits good safety in healthy adults, similar to BCG
	NCT02729571	Ib	2015.9	2018.3	Newborn infants	54	i.d.	Safety and reactogenicity in infants and adults	MTBVAC has good safety and immunogenicity, and can induce long-lasting CD4 cell responses in infants
	NCT02933281	IIa	2018.5	2021.9	Healthy adults	144	i.d.	Safety and reactogenicity of MTBVAC at escalating dose levels compared to BCG vaccine by assessing number of participants with AEs and SAEs	NA
	NCT03536117	IIa	2019.2	2022.5	Newborn infants	99	i.d.	1. Number of participants with treatment-related AEs as defined in protocol 2. Immunogenicity analysis in infants	NA
	NCT04975178	III	2022.9	2029.9	Newborn infants	7120	i.d.	Prevention of TB disease in healthy HIV-uninfected and HIV-exposed uninfected neonates	NA
	NCT06272812	IIb	2024.9	2028.3	Adolescents and Adults	4300	i.d.	To evaluate the protective efficacy of MTBVAC against bacteriologically confirmed pulmonary TB disease, diagnosed by more than one diagnostic test with sputum obtained before initiation of TB treatment as compared to placebo	NA
**M72/AS01E**	NCT01755598	IIb	2014.8	2018.11	LTBI individuals	3575	i.m.	Incident rates of definite PTB disease, not associated with HIV-infection, meeting the case definition	LTBI population vaccinated with M72/AS01E can prevent latent infections from developing into TB, the vaccine efficacy in the 36th month was 49.7%
	NCT04556981	II	2020.11	2022.8	HIV-positive patients	402	i.m.	Number of subjects with solicited local symptoms and solicited general symptoms, unsolicited AEs and SAEs, different levels biochemical and hematological levels	NA
	NCT06062238	III	2024.3	2028.4	Children, Adults	20,000	i.m.	Number of participants with solicited AEs, unsolicited AEs, and SAEs	NA
**GamTBvac**	NCT03255278	I	2017.1	2017.12	Healthy adults	60	s.c.	The number of AEs	Different doses of vaccines were evaluated for immunogenicity, with half dose (0.5 mL) having the best effect
	NCT03878004	II	2018.12	2020.5	Healthy adults	180	s.c.	1. Level of IFN-γ secretion in whole blood or PBMC fraction 2. Number of participants with AEs	The vaccine is well tolerated and induces specific and persistent Th1 and Humoral immunity responses
	NCT04975737	III	2022.1	2025.11	Healthy adults	7180	s.c.	Preventive efficacy (Ep)	NA
**ID93 + GLA-SE**	NCT01599897	I	2012.8	2014.5	Healthy adults	60	i.m.	Number of patients experiencing AEs	Showing a satisfactory safety profile and eliciting a functional humoral and T-helper 1 type cellular response
	NCT06670755	I	2024.12	2027.7	Healthy adults	48	i.m.	Safety of BCG challenge by the aerosol inhaled route in healthy volunteers and recently ID93/GLA-SE-vaccinated adult volunteers	NA
	NCT06714513	I	2024.12	2026.12	Adults, older Adults	144	i.m.	1. Safety 2. Immunogenicity	NA
	NCT02465216	IIa	2015.6	2017.1	Healthy adults	60	i.m.	Number of AEs	The antigen-specific IgG and CD4 T cell responses induced by a dose of 2 µg ID93 + 5 µg GLA-SE were significantly higher than those induced by placebo, and lasted for 6 months
**H107e/CAF10b**	NCT06050356	I	2024.3	2026.11	Healthy adults	140	i.m.	1. Frequencies of H107e-specific IFN-γ producing T cells before first i.m. vaccination and two weeks after the second i.m. vaccination 2. Frequencies of BCG-specific T-cells producing IFN-γ and/or IL-17 induced by H107e/CAF^®^10b + BCG vs. BCG alone	NA
**AEC/BC02**	NCT03026972	Ia	2018.4	2019.10	Healthy adults	25	i.m.	The number of participants with Adverse Events after coxal muscle injection	NA
	NCT04239313	Ib	2020.5	2022.6	Healthy adults	30	i.m.	The number of AEs after i.m.	NA
**ChAdOx1.85A + MVA85A**	NCT01829490	I	2013.7	2016.4	Healthy adults	42	i.m.	Safety of ChAdOx1 85A vaccination with and without MVA85A boost vaccination in healthy, BCG-vaccinated adults	ChAdOx1.85A induces Ag85A specific CD4 + T and CD8 + T cell responses
	NCT03681860	IIa	2019.7	2021.5	Healthy adults	72	i.m.	Safety and immunogenicity	NA
**AdHu5Ag85A**	NCT02337270	I	2017.9	2021.9	Healthy adults	36	Aerosol	Number of participants reporting AEs	Compared with i.m., aerosol delivered AdHu5Ag85A vaccine has advantages in inducing respiratory epithelium immunity
**TB/FLU-05E**	NCT05945498	I	2023.5	2023.9	Healthy adults	51	Intranasal Injection	Number of participants with local and AEs and SAEs	NA
**BNT164**	NCT05537038	I	2023.4	2025.12	Healthy adults	120	i.m.	1. Frequency of solicited local reactions (pain, erythema/redness, induration/swelling) at the injection site up to 7 days after each dose 2. Frequency of solicited systemic reactions (vomiting, diarrhea, headache, fatigue, muscle pain and joint pain, chills, and fever) up to 7 days after each dose 3. Proportion of participants with at least one AE occurring from each dose to 28 days after each dose 4. Proportion of participants with at least one AE occurring from Dose 1 to 28 days post-Dose 3 5. Proportion of participants with at least one SAE or medically attended adverse event (MAAE) occurring from Dose 1 up to 168 days post-Dose 3 6. Number of AEs from Dose 1 to 28 days post-Dose 3	NA
	NCT05547464	IIa	2023.7	2027.5	Healthy adults	732	i.m.	1. Frequency of solicited local reactions (pain, erythema/redness, induration/swelling) at the injection site up to 7 days after each dose 2. Frequency of solicited systemic reactions (vomiting, diarrhea, headache, fatigue, muscle pain and joint pain, chills, and fever) up to 7 days after each dose 3. Proportion of participants with at least one AE occurring from each dose to 28 days after each dose 4. Proportion of participants with at least one SAE or AE of special interest occurring from Dose 1 up to 168 days post Dose 3 5. Proportion of participants with at least one SAE or medically attended adverse event (MAAE) occurring from Dose 1 up to 168 days post-Dose 3 6. Number of unsolicited AEs from Dose 1 to 28 days post-Dose 3	NA

Abbreviations: i.m., intramuscular injection; s.c., subcutaneous injection; i.d., intradermal injection; AEs: adverse events; SAEs: serious adverse events; NA, not available.

## Data Availability

No new data were created or analyzed in this study. Data sharing is not applicable to this article.

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
