# Peer review of "Recent Advances in Clinical Research of Prophylactic Vaccines Against Tuberculosis"

_vaccines, 2025, doi:10.3390/vaccines13090959_

Round 1
Reviewer 1 Report
Comments and Suggestions for Authors
This is a well written review on a subject of ever increasing importance.
Author Response
Comments: This is a well written review on a subject of ever increasing importance.
Response: Thank you very much for your positive feedback on the review. We deeply appreciate your recognition of the review’s writing quality and its focus on a subject of growing importance.
Reviewer 2 Report
Comments and Suggestions for Authors
The article titled: Recent Advances in Clinical Research of Prophylactic Tuberculosis Vaccines is a comprehensive summary of the development of preventive tuberculosis vaccines, because the use of Bacille Calmette-Guerin (BCG) vaccine is not sufficient in the reduction of TB transmission. Tuberculosis is still the most important infectious disease, killing more people than malaria and AIDS combined.
Recombinant live attenuated MTB vaccines are still in clinical trials, specifically in phases Ia., and IIb. showed encouraging effects in children with fewer side effects compared to BCG. Using recombinant subunit vaccines, viral vector vaccines, and mRNA platforms are all encouraging efforts to prevent the spread of TB.
The article is well-designed and proportionate, including the results of clinical trials on vaccine development.
The article is highly recommended for publication in the present form.
Author Response
Thank you sincerely for your thorough review and highly positive evaluation of our article. Your recognition of the article as a "comprehensive summary of the development of preventive tuberculosis (TB) vaccines" and your affirmation of its well-designed structure, proportionate content, and inclusion of critical clinical trial results mean a great deal to our research team—it not only validates the rigor and value of our work but also reinforces our confidence in the significance of this topic.
Reviewer 3 Report
Comments and Suggestions for Authors
This review did a great overview of the current TB vaccine candidates. It will be a excellent resource for vaccine researchers. However, to be precise, I have a few comment below:
- The authors used "prophylactic" in the title. This is not accurate. Some vaccines are trialed on healthy/QFN- for prevention, while some are trialed on LTBI/QFN+ to stop reactivation. So it will be great if the authors could clarify the target population in each candidate. Also correct the title.
- For some vaccines, the administration route is not clear in the manuscript.
- It will be great to emphasize that "M72/AS01E" is the only one with a positive efficacy readout so far.
- Lastly, the authors did a good job summarizing all the vaccines in a table. There is a "Start time". Is it possible to add a potential "End time" or timeline?
Author Response
Comments 1: The authors used "prophylactic" in the title. This is not accurate. Some vaccines are trialed on healthy/QFN- for prevention, while some are trialed on LTBI/QFN+ to stop reactivation. So it will be great if the authors could clarify the target population in each candidate. Also correct the title.
Response 1: Thank you for pointing this out. We agree with this comment. The title has been revised.
Comments 2: For some vaccines, the administration route is not clear in the manuscript.
Response 2: We agree with your comment. The administration routes of the vaccines have been added to Table 1.
Comments 3: It will be great to emphasize that "M72/AS01E" is the only one with a positive efficacy readout so far.
Response 3: Thank you for pointing this out. We agree with this comment. Emphasis has been added at Line 131.
Comments 4: Lastly, the authors did a good job summarizing all the vaccines in a table. There is a "Start time". Is it possible to add a potential "End time" or timeline?
Response 4:We agree with your comment. The end time has been added to Table 1.
Reviewer 4 Report
Comments and Suggestions for Authors
The manuscript provides a comprehensive and well-structured review of current clinical progress in prophylactic TB vaccines, covering live-attenuated, recombinant subunit, viral-vector, and emerging mRNA candidates. It summarises key trial outcomes, safety profiles, immunogenicity data, and future directions. The article is timely, as TB remains the leading cause of death from a single infectious agent, and vaccine innovation is a global priority.
The review demonstrates breadth and depth of coverage, citing major vaccine candidates under investigation and discussing novel strategies such as antigen optimization, adjuvant development, and biomarker identification. The narrative is clear and scientifically rigorous.
Furthermore, the manuscript demonstrates strong comprehensiveness by covering all major categories of preventive TB vaccines, including live attenuated, subunit, viral vector, and mRNA platforms. It effectively summarises phase I–III clinical trial data, providing quantitative details on sample sizes, immunogenicity, adverse events, and efficacy, and offering readers a clear understanding of the current evidence base. Furthermore, the review provides valuable forward-looking insights by discussing key considerations, including clinical trial endpoints, potential diagnostic interferences, the development of novel adjuvants, and the identification of biomarkers. Its timeliness is also notable, as it highlights emerging vaccine platforms such as mRNA and bioinformatics-driven antigen design, underscoring the manuscript’s relevance to current global TB vaccine research and development efforts.
Abstract: Excellent summary; consider emphasizing the novelty of including mRNA vaccines.
Live Attenuated Vaccines: Well-detailed. It would benefit from a concise table summarising trial phases, populations, and outcomes.
Recombinant Subunit Vaccines: Excellent detail on M72/AS01E. Emphasize its significance as the only subunit vaccine with demonstrated efficacy in a Phase IIb trial.
mRNA Vaccines: Brief; should include more context on why mRNA might overcome limitations of protein-based vaccines (e.g., speed of design, induction of both T and B cell responses).
Novel Strategies: Well written, but could highlight host-directed therapies and combined vaccine–therapeutic approaches.
Author Response
Thank you very much for your positive feedback on the review. We deeply appreciate your recognition of the review’s writing quality and its focus on a subject of growing importance.